# Insect and Plant Diversity in Hot-Spring Ecosystems during the Jurassic-Cretaceous Boundary from Spain (Aguilar Fm., Palencia)

**DOI:** 10.3390/biology11020273

**Published:** 2022-02-09

**Authors:** Artai A. Santos, André Nel, Iván Rodríguez-Barreiro, Luis M. Sender, Torsten Wappler, José B. Diez

**Affiliations:** 1Centro de Investigación Mariña, Universidade de Vigo (CIM-UVIGO), 36310 Vigo, Spain; asantos@uvigo.es (A.A.S.); ivan.rodriguez.barreiro@uvigo.es (I.R.-B.); 2Departamento de Xeociencias Mariñas e Ordenación do Territorio, Facultade de Ciencias do Mar, Universidade de Vigo, 36310 Vigo, Spain; 3Institut de Systématique, Évolution, Biodiversité (ISYEB), Muséum National d’Histoire Naturelle, CNRS, Sorbonne Université, EPHE, Université des Antilles, CP50, 57 Rue Cuvier, 75005 Paris, France; andre.nel@mnhn.fr; 4Área de Paleontología, Facultad de Ciencias. Edificio C, Universidad de Zaragoza, 50009 Zaragoza, Spain; luismisender@gmail.com; 5Department of Natural History, Hessisches Landesmuseum Darmstadt, Friedensplatz 1, 64283 Darmstadt, Germany; torsten.wappler@hlmd.de; 6Paleontology Section, Institute of Geosciences, Rheinische Friedrich-Wilhelms Universität Bonn, 53115 Bonn, Germany

**Keywords:** travertines, Tithonian, Berriasian, hydrothermalism, Bennettitales, Odonata, migrations, Iberian Peninsula

## Abstract

**Simple Summary:**

In this study, we show and identify new plant and insect remains found in travertine deposits from the Jurassic-Cretaceous boundary of the Aguilar Formation (Palencia, North Spain). From this hot-spring palaeoenvironment, we have identified the presence of dragonflies of the families Cymatophlebiidae and Aktassiidae, representing the first report of these families for the Iberian Peninsula. In addition, we find a flora dominated by Bennettitales and the presence of ferns that differ from other floras of the same age and geographical area. The unusual environmental and palaeoecological conditions of this hot-spring environment are also discussed, suggesting that this niche was an “ecological oasis” for some plants and insects.

**Abstract:**

Hydrothermal palaeoenvironments are very uncommon in Upper Jurassic and Lower Cretaceous deposits worldwide. We present new plant and insect remains from travertines formed during the Jurassic-Cretaceous boundary in northern Spain (Aguilar Fm., Palencia province). A total of 136 plant specimens and three insect wings were collected and studied. This entomofauna consists of dragonfly (Odonata) wings including Cymatophlebiidae and an undetermined new genus and species of Aktassiidae, representing the first report of these families for the Iberian Peninsula. The fossil flora shows different morphotypes of plants, which have been tentatively assigned to three different genera. The taphocoenosis of the flora was dominated by Bennettitales (98.5%) including cf. *Pterophyllum* sp., *Ptilophyllum* cf. *acutifolium*, *Ptilophyllum* cf. *pecten*, *Ptilophyllum* cf. *pectiniformis* and cf. *Ptilophyllum* sp., and the occasional presence of ferns (1.5%) represented by the taxon *Cladophlebis* cf. *denticulata*. The presence of the Anisoptera *Cymatophlebia* cf. *longialata* suggests a higher affinity for a Tithonian age of the studied site, and the anatomy and palaeogeographical distribution of this species suggest capacity to migrate for rather long distances. The floristic composition of the site differs remarkably from other Tithonian-Berriasian floras of the Iberian Peninsula. The presence of Odonata and the distinctive flora in (semi)arid conditions could be due to the hot-spring providing an environmental niche with constant conditions of warmth and humidity forming an ‘ecological oasis’.

## 1. Introduction

Plants and insects are two of the most important ecological groups in continental ecosystems. Currently, they represent more than 80% of the biodiversity and biomass of the planet [1,2,3,4,5]. In addition, the evolutionary co-association between these two groups has been widely studied in modern and ancient ecosystems. Hot springs represent unique ecosystems with distinctive plant, animal, fungal, and bacterial communities [6,7,8] due to the stable temperature and the chemical composition, unusual pH values and concentrations of solids [9,10,11]. Hydrothermal sites could represent an environmental refuge for animals and plants with constant temperature and humidity needs [8,12,13,14]. In current ecosystems, some insects such as different species of dragonflies and damselflies are known to establish local populations in warm springs [8,15,16].

If Holocene travertine fossil deposits are frequent, their occurrences become increasingly scarcer in the Neogene, Paleogene, and overall in the Mesozoic, probably in relation to the brittleness of these sediments. Examples of these types of deposits during the Jurassic and Lower Cretaceous are few and mostly located in America (see Table 1). In Spain, in addition to the present case study [17,18,19,20,21,22,23], it is only worth highlighting an oncolithic-rich travertine deposit associated to springs and ephemeral lakes from the Kimmeridgian of Asturias (NW Spain), in the Vega Fm. [24,25,26]. Evidence of roots and wood remains, charophytes, and ostracods were found in this outcrop, but no insects or identifiable plant remains occur there. In America the Middle-Late Jurassic deposits of hot spring environments from Patagonia, Argentina stands out. This hot-springs system was widely studied; plants, arthropods, microbial, and fungal remains were identified [6,7,27,28,29,30,31,32,33]. In addition, plant remains have also been found in other hot-spring deposits of Jurassic age in different parts of North America (See Table 1) [34,35,36,37]. If fossil insects are rather frequent in Holocene travertine deposits, these are much rarer in the deeper past. Among noticeable exceptions is the Eocene outcrop of Sézanne (France) that has given a rich flora and few well preserved arthropods [38,39], and the rich Miocene entomofauna of the ‘Böttinger Marmor’ in Southwestern Germany [40,41,42].

The present study is focused on new fossil remains of insects and plants from hot-spring deposits, that were formed during the Jurassic-Cretaceous boundary in a geothermal spring environment under (semi)arid conditions [6,7,43]. This study also provides the first Tithonian-Berriasian insect remains from the Iberian Peninsula and one of the few hot-springs deposits with plant and insect fossils of the world. On the other hand, the Tithonian-Berriasian floras are scarce in Europe, and they are really unusual in the Iberian Peninsula. In Portugal, the floras of this age are restricted to micro and meso-fossils [44,45,46]. While in Spain, there are only three references to the flora of this age: (1) Early Berriasian macro and microflora from the Villar del Arzobispo Fm. in the Galve area (Eastern Spain) [47], deposited in a deltaic environment and mainly dominated by pteridophytes; (2) Berriasian to Barremian? gymnosperms and pteridophytes in lacustrine deposits from Sierra de Montsec (NE Spain) [48]; and (3) Tithonian-Berriasian leaves of Bennettitales and ferns from the Aguilar Fm. (North of Spain) [23], which were deposited in a hot spring environment located relatively close and laterally equivalent to the new site. The flora studied by Diéguez et al. [23] was collected from the same levels of travertine in the Aguilar Fm. that were studied in the present work. However, according to their map location of the fossil site, just a few hundred meters from ours towards the SW. As sampling was carried out before 2009, it seems that the recent construction of the A-67 highway prevents to accessing this site.

Hot springs are commonly associated with active volcanic regions [49], as was the case in the area studied. In this case, the hot water circulated and emerged to the surface through an active fault at the boundary of a large wetland system during the Tithonian-Berriasian in northern Spain [23]. In this hot-spring complex, a flora dominated by Bennettitales was established promoted by the humid conditions and warm temperatures. The fossil record also shows that soon the insects took advantage of this new ecosystem niche.

On the other hand, fossil evidence of Jurassic insects in the Iberian Peninsula is non-existent. The only reference to Jurassic insects of this age is some Middle Jurassic (Aalenian) plant-insect interactions at El Pedregal Fm. from the Sierra de Javalambre (Teruel, Spain) [50]. Only the Sierra del Montsec (Lleida, Spain) sites are known at the Lower Cretaceous base, whose age ranges between the late Berriasian? and Barremian. In this area, remains of insects of several orders occur, including, Orthoptera, Odonata, Hemiptera, Hymenoptera, Trichoptera, Blattodea (including Isoptera), Diptera, Neuroptera, Coleoptera, Ephemeroptera, Phasmatodea, and Raphidioptera [51,52,53,54,55,56,57,58].

Hence, the main objectives of this study are to (1) identify the palaeodiversity of plants and animals in a new hot-spring site from the Tithonian-Berriasian of Spain, (2) interpret the biological and ecological meaning of this palaeoecosystem, and (3) deduce the palaeoenvironmental and palaeogeographical implications of these fossils.

## 2. Geographic and Geological Setting

The fossil site is located in an outcrop near the road that connects the villages of Villela and Becerril del Carpio in the province of Palencia (northwest of Spain; see Figure 1), where this zone is integrated in the “Las Loras UNESCO Geopark”. The fossils have been found in materials from the base of the Santa María la Real Member, corresponding to the lower part of the Aguilar Fm. within the Campóo Group of Upper Tithonian-Berriasian age.

The deposits containing the studied fossils are related to the tectonic-sedimentary activity during the Mesozoic in this area. The beginning of the rifting of the Bay of Biscay during the Late Jurassic-Early Cretaceous had several implications for the palaeogeography, hydrography, and sedimentation in Northern Spain. Including the deposition of the alluvial-palustrine Aguilar Basin [22,59,60,61]. The sedimentation was controlled mainly by two significant syn-sedimentary faults: the Villela Fault (also known as Becerril Fault), which delimits the Western margin of the Aguilar Basin, and the Ubierna Fault. Faulting resulted in abrupt changes of both facies and their thickness. Furthermore, numerous minor faults also affected the distribution of the materials, promoting local differences through the basin [22,59,61].

The Aguilar Basin is divided into three stratigraphic groups according to different tectonic sequences where the older one (the Campóo Group) was sub-divided originally into four formations first [59] but, more recently, a revision reduced them to two, the Aguilar and Frontada formations [21]. The Aguilar Fm. is the thickest and more complex unit within the Campóo Group. It comprises three lithosomes of lacustrine/palustrine limestones and three lithozones of fluvial and alluvial terrigenous sediments. In this formation, lateral changes of facies are notable, affecting the proportion between calcareous and terrigenous sediments. The closer to the depocenter, the higher the calcareous materials [21]. Taking this into account, the latter authors made the following sub-division of the Aguilar Fm. according to each of the lithosomes: the palustrine/lacustrine calcareous Santa María la Real, Prandillera, and La Presa members; and the fluvial/alluvial terrigenous Becerril, Hormiguera, and Barcenilla members.

The Santa María la Real Member, target in this study, represents the older and thicker of the three calcareous members. It has a maximum thickness of 250 m, and it is composed of metric sequences formed by the following facies: (1) dark marls rich in organic matter with the presence of black pebbles, (2) nodular marly limestones, and (3) micritic limestones with charophytes and ostracods (Figure 1B) [21].

The environmental interpretation for the Santa María la Real Mb. consists of a lacustrine-palustrine system with frequent but slight variations on the water level, these being more prominent on the margins of the basin within semi-arid conditions [21,22,23,62]. More locally, in the studied area, hot-springs deposits containing fossil plant remains and related to a hydrothermal source associated with the Villela Fault were located on the Western margin of the Aguilar lacustrine/palustrine system. The fossil remains were found in palustrine limestones with channel-shaped structures where concentrically laminated spherical oncolithic levels occur, and vegetation remains are coated by bacterial and cyanobacterial growth [20,23]. Due to the facies observed, this association is interpreted by Hernández et al. [20] as a fossil travertine.

The age of the levels where the fossils were found would be late Tithonian–early Berriasian according to the biostratigraphic works made in the Santa María de la Real Mb. The charophyte assemblage found by Hernández et al. [21] in this member corresponds to the *Maillardii* biozone, which has an interval of a late Tithonian–early Middle Berriasian age. Additionally, Schudack [19] in his study of ostracods, found *Clavator reidii* var. *pseudoglobatoroides* at the top of the Santa María de la Real Mb., whose presence indicates that the upper part of the member cannot be older than early Berriasian.

During the Tithonian-Berriasian, the study area was occupied by a shallow lake system, developing wetlands, and colonization by charophytes and subsequent desiccation [43], in which carbonate hydrothermal contributions would take place with the development of covers bacterial [20]. This data would constitute a marked stresses environment due to the abundance and variety of leaves of cycadophytes related to dry environments [23]. The presence of different morphotypes of rhizolites reflects changes in the substrate, from humidification conditions to desiccation, suggesting correlative changes in the humidity/aridity conditions in the Aguilar Fm. [63]. In this sense, Pujalte et al. [43] indicated that the presence of evaporitic deposits laterally correlated to the limestones would be indicative of the existence of (semi)arid conditions, which would be reinforced by the xerophytic fossil flora assemblage that was found in this area [23].

## 3. Material and Methods

The fossil assemblage consists of three insect fossils (Figure 2) and 136 samples of plants (Figures 3 and 4) recovered on a fieldtrip during the 2021 summer in a new travertine deposit located in Becerril del Carpio (Palencia, Spain). The Becerril fossil macroflora is dominated by leaves of cycadophytes—putative Bennettitales—with the occasional presence of ferns. The entomofauna consists of wings of Odonata. The fossils were prepared following standard techniques in the Department of Geosciences of Vigo University, using a micro-pneumatic hammer and sharp needles. Selected fossils were photographed in detail using a Canon EOS50D with a 60-mm macro lens and a Leica MC170-HD Camera attached to a Leica M205-C stereomicroscope. Some samples were photographed submerged in distilled H_2_O to improve image contrast and quality. The fossils will be stored at the Las Loras UNESCO Geopark.

We have tried to carry out a palynological analysis of the levels studied. The samples were analyzed using standard palynological techniques at the Palynology Laboratory of the Department of Marine Geosciences (University of Vigo). The three studied samples from the travertine deposit were negative in palynomorphs. Only some phytoclasts, wood remains, and cuticle fragments were found. The small size and bad preservation of the cuticles remains from the palynological samples prevents the identification or assignation to any group of plants.

Abbreviations used in the text and figures: AA = analis anterior, Ax1 and Ax2 = primary antenodal crossveins, CuA = cubitus anterior, CuAa = anterior branch of CuA, CuAb = posterior branch of CuA, CuP = cubitus posterior, h = hypertriangle, IR1 and IR2 intercalary convex veins between branches of RP, MA = media anterior, MAa = convex anterior branch of MA, MAb = posterior branch of MA, MP = concave media posterior, Mspl = intercalary concave longitudinal vein between MAa and MP, PsA = pseudo-anal vein between AA + CuP and MP + CuA, Pt = pterostigma, RA = convex radius anterior, RP = concave radius posterior; RP2 and RP3/4 = branches of RP, Rspl = intercalary concave longitudinal vein between RP2 and IR2, ScP = subcosta posterior, t = discoidal triangle.

## 4. Systematic Palaeontology

### 4.1. Insects

Order Odonata Fabricius, 1793

Suborder Anisoptera Selys in Selys and Hagen, 1854

Clade Aeshnoptera Bechly, 1996

Family Cymatophlebiidae Handlirsch, 1906 (*sensu* Bechly et al., 2001)

Subfamily Cymatophlebiinae Handlirsch, 1906 (*sensu* Bechly et al., 2001)

Genus *Cymatophlebia* Deichmüller, 1886

Type species. *Cymatophlebia longialata* (Münster in Germar, 1839) (early Tithonian, Germany).

Other species. *Cymatophlebia suevica* Bechly et al., 2001 (early Kimmeridgian, Germany), *Cymatophlebia herrlenae* Bechly et al., 2001 (early Kimmeridgian, Germany), *Cymatophlebia densa* Bechly, 2001 (early Tithonian, Germany), *Cymatophlebia pumilio* Bechly et al., 2001 (early Tithonian, Germany), *Cymatophlebia kuempeli* Bechly et al., 2001 (early Tithonian, Germany), *Cymatophlebia purbeckensis* Bechly et al., 2001 (Early Cretaceous, UK), *Cymatophlebia standingae* (Jarzembowski, 1994) (Early Cretaceous, UK), *Cymatophlebia zdrzaleki* (Jarzembowski, 1994) (Early Cretaceous, UK), *Cymatophlebia yixianensis* Zheng et al., 2018 (Early Cretaceous, China). Bechly et al. (2001) considered ‘*Cymatophlebia’ mongolica* Cockerell, 1924 as a nomen dubium in Anisoptera incertae sedis.

*Cymatophlebia* cf. *longialata* (Münster in Germar, 1839)

(Figure 2A–C)

Material: Specimen AG-I-01 (a nearly complete hind wing with only apical part distal of level of pterostigma missing), specimen AG-I-03 (a fragment of wing), both stored at Las Loras UNESCO Geopark.

Age and outcrop: Tithonian-Berriasian (Jurassic-Cretaceous boundary), Aguilar Fm. (Becerril del Carpio, Palencia, Spain).

Description: Preserved part of hind wing 55.5 mm long; width at nodus 19.2 mm; distance from base to nodus 29.1 mm; distance from nodus to pterostigma 22.8 mm. Ax2 is slightly distal of level of discoidal triangle; Ax1 not preserved; ca. 2–3 antenodal crossveins between Ax1 and Ax2 and eight antenodal crossveins distal of Ax2; two rows of secondary antenodal crossveins not aligned; ca. 16 postnodal crossveins between nodus and pterostigma; postnodal crossveins and postsubnodal crossveins not aligned; no pseudo-ScP distal of nodus; pterostigma incomplete, braced; RP1 and RP2 run parallel up to pterostigma with two rows of cells in-between except for five most basal cells; up to three rows of cells between undulated parts of RP2 and IR2; four rows of cells between IR2 and curved Rspl; three oblique crossveins ‘O’ between RP2 and IR2, two cells, six cells and seven cells distal of subnodus; two oblique secondary veins between IR2 and RP3/4 immediately basal of origin of Rspl; RP3/4 and MAa distinctly undulated, basally closely parallel, but strongly divergent near wing margin; distinct Mspl present, as well-defined as Rspl, curved, and with up to three rows of cells between Mspl and MAa; four rows of cells in basal part of postdiscoidal area (width near discoidal triangle 4.2 mm; width at wing margin 7.8 mm); discoidal triangle divided into five cells; length of anterior side 6.9 mm; of basal side 2.8 mm; of straight distal side MAb 7.2 mm; hypertriangle divided by two crossveins (length 8.3 mm); subdiscoidal triangle divided into three cells; no cubito-anal crossvein in submedian space between CuP-crossing and PsA; 10 posterior branches of CuA (including CuAb); up to 10 or 11 rows of cells in cubito-anal area between CuAa and posterior wing margin; PsA angled; anal loop small, divided into six cells, and posteriorly rather well-closed (by a composite crossvein between CuAb and AA1b, or by a fusion of CuAb with AA1b?); four posterior branches of AA between CuA and anal margin; anal margin rounded without an anal angle or anal triangle (female specimen). About 12 or 13 rows of cells in anal area.

Specimen AG-I-03. A fragment of a wing showing radial area in its part below pterostigma and distal part of MAa. All visible characters identical to those of specimen AG-I-01.

Remarks: Specimen AG-I-01. This hind wing has the characters diagnostic of the Cymatophlebiinae, viz. dense wing venation with numerous cells; two rows of secondary antenodal crossveins not aligned; postnodal crossveins and postsubnodal crossveins not aligned; pterostigma elongated and braced; apparent furcation of AA into an anterior secondary branch PsA and a posterior main branch AAa; Rspl well-defined and curved with several rows of cells between it and IR2; Mspl curved; MA, RP3/4, IR2 and RP2 strongly undulated; several rows of cells in areas between MAa and RP3/4, and between IR2 and RP2 along posterior wing margin; MAa and RP3/4 reach posterior wing margin at right angles; two oblique secondary veins between IR2 and RP3/4 immediately basal of origin of Rspl; two or three oblique veins ‘O’; discoidal triangle divided into several cells; anal and cubito-anal areas very wide in hind wing; CuAa with numerous posterior branches; anal loop reduced. It has also the characters of the genus *Cymatophlebia*, viz. no accessory cubito-antenodal crossveins between CuP-crossing and PsA, postdiscoidal area hardly widened distally; only few secondary antenodal crossveins between Axl and Ax2, subdiscoidal triangle three-celled; more than three rows of cells in basal postdiscoidal area of hind wing; wing longer than 70 mm, very dense venation with numerous cells; well-defined and curved Mspl, convex pseudo-veins in postdiscoidal area (also between Mspl and MA).

Specimen AG-I-01 differs from *Cymatophlebia pumilio* in the presence of three cells in the subdiscoidal triangle, larger wing (ca. 58 mm long and 19.2 mm wide at nodus, instead of 46–49 mm long and 14 mm wide at nodus). *Cymatophlebia standingae* has hind wing more than 77 mm long, instead of ca. 58 mm in specimen AG-I-01, and five rows of cells in area between IR2 and Rspl, instead of four in specimen AG-I-01). *Cymatophlebia suevica* has five rows of cells in basal part of postdiscoidal area and six rows of cells between MAa and Mspl. *Cymatophlebia herrlenae* is known from forewings only, hardly comparable to specimen AG-I-01; its Mspl is very weak in contrast with in specimen AG-I-01, but this character is rather a forewing one in this genus. Nevertheless, its wings are larger than that of specimen AG-I-01 (ca. 65 mm long). *Cymatophlebia purbeckensis* is also based on an incomplete forewing, hardly comparable to the hind wing of specimen AG-I-01 and its alleged wing length is ca. 55 mm. It has only two rows of cells between MAa and Mspl, instead of three in specimen AG-I-01. *Cymatophlebia zdrzaleki* differs from specimen AG-I-01 in the presence of four rows of cells between RP1 and RP2 and of six rows of cells between RP2 and IR2 below the pterostigma. *Cymatophlebia densa* differs from specimen AG-I-01 in the presence of 5–6 rows of cells between IR2 and Rspl, and three rows of cells between RP1 and RP2 basal of pterostigma. *Cymatophlebia yixianensis* has three rows of cells between IR2 and Rspl as in specimen AG-I-01, but three rows of cells between RP1 and RP2. Its Mspl is very poorly defined, but its holotype is a forewing in which Mspl is generally less well-defined than in the hind wings of the *Cymatophlebia* spp.

*Cymatophlebia kuempeli* and *Cymatophlebia longialata* are very similar. They differ in characters not observable in specimen AG-I-01. *Cymatophlebia kuempeli* has a hind wing 64.0 mm long and 18.7 mm wide at nodus, thus probably larger than that of specimen AG-I-01. In addition, its hind wing vein Mspl is poorly defined differently from that of specimen AG-I-01, and it has only three rows of cells between RP3/4 and MAa along posterior wing margin, instead of 5–6 in specimen AG-I-01. *Cymatophlebia longialata* has three to six rows of cells between RP3/4 and MAa along posterior wing margin (thus this character is somewhat variable), and generally a poorly defined Mspl in hind wing. However, some hind wings can have a better defined Mspl than others (e.g., specimen JME SOS 1715, see Bechly et al. [64]: text-figure 46). The wing size and proportions of specimen AG-I-01 fit well in the range of the hind wings of *Cymatophlebia longialata*.

Clade Petalurida Bechly, 1996

Family? Aktassiidae Pritykina, 1968 (*sensu* Nel et al. [65])

New genus and species undetermined

(Figure 2D)

Material: Specimen AG-I-02

Age and outcrop: Tithonian-Berriasian (Jurassic-Cretaceous boundary), Aguilar Fm. (Becerril del Carpio, Palencia, Spain).

Description: A wing fragment showing distal parts of RP2, IR2, RP3/4, and MAa, 24.3 mm long, 13.1 mm wide; RP2 and IR2 closely parallel, basally weakly curved with one row of cells in-between, IR2 distally with an anterior curvature and two rows of cell between it and RP2; Rspl very poorly defined, nearly absent, with three rows of cells between it and IR2; area between IR2 and RP3/4 very broad with three longitudinal veins; RP3/4 nearly straight; MAa curved; area between RP3/4 and MAa strongly broadened in its distal part, with 13 rows of cells in-between along posterior wing margin and three longitudinal veinlets; postdiscoidal area probably very broad after its preserved part.

Remarks: This wing fragment strongly differs from the specimens AG-I-01 and AG-I-03 in the very broad area between RP3/4 and MAa and the strongly reduced Rspl. Such a broad area between RP3/4 and MAa is quite uncommon among the Odonata, only found in some Petalurida: Aktassiidae such as *Aeshnogomphus buchi* Hagen [66] or *Pseudocymatophlebia hennigi* Nel et al. [65]. Even the *Aktassia* spp. have a smaller area between these veins [65]. The Aeshnoptera and the Aeschnidiidae that can have large areas between IR2 and RP3/4 do not share so broad areas between RP3/4 and IR2 [64,67]. *Aeshnogomphus* differs from the new fossil in the absence of distal curvature of IR2, a broader area between IR2 and RP3/4 and a narrower one between RP3/4 and MAa. *Pseudocymatophlebia* better fits with the new fossil in the very broad area between RP3/4 and MAa but it also has no curvature of IR2, and its RP3/4 is weakly but distinctly undulate instead of being straight.

Thus, the new fossil corresponds to a new genus and species, probably belonging to the Aktassiidae. However, it is too incomplete to allow the description of a new taxon.

**Figure 2 biology-11-00273-f002:**
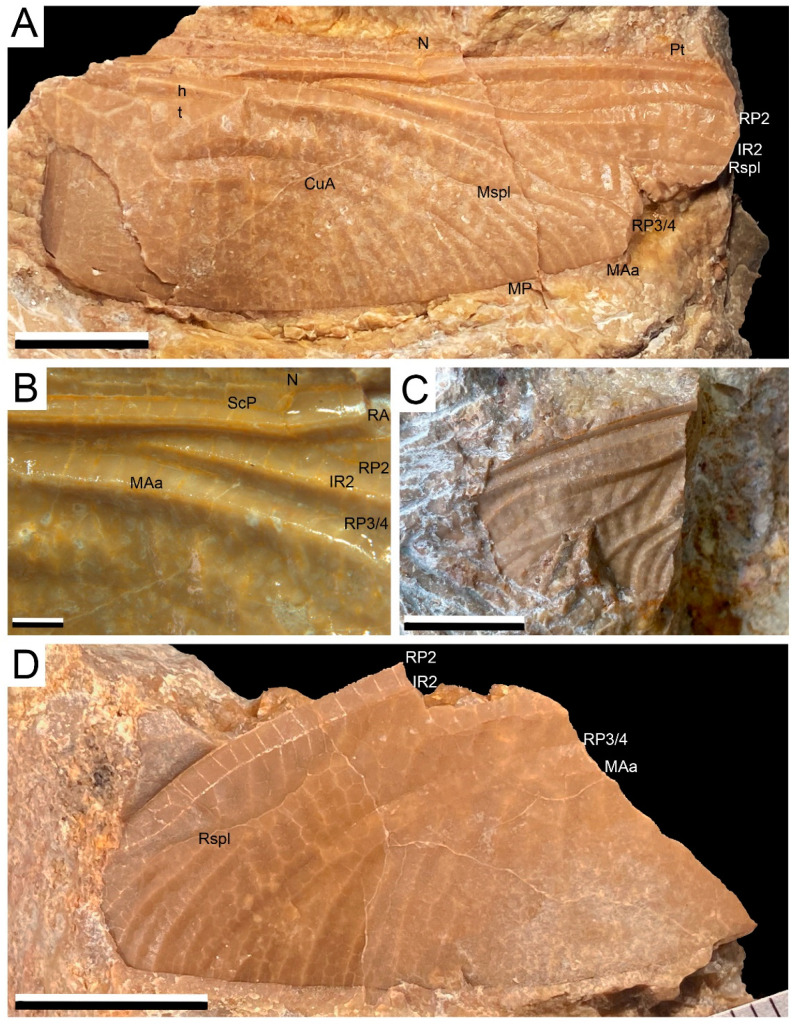
Dragonfly wings from the Becerril section: (**A**–**C**). Fossil remains of *Cymatophlebia* cf. *longialata* indicating venational features, AG-I-01 (**A**,**B**) and AG-I-02 (**C**); (**D**) Wing of a new undetermined genus and species of Aktassiidae, AG-I-03.

### 4.2. Plants

The remains of plants consist of fragments of leaves that occur as impressions with very occasional carbonaceous thin crusts (Figure 3 and Figure 4). No cuticular remains could be recovered due to preservation conditions, and samples selected for SEM analyses do not allow the observation of histological characteristics of the specimens. In some cases, these characters are essential, and their absence makes their accurate identification difficult. However, in the case of the leaves of gymnosperms both the structure of the base of leaflets and the insertion of them in the rachis indicate their relation of these remains to several genera within Order Bennettitales.

Class Polypodiopsida

Order Filicales

Family incertae sedis

Genus *Cladophlebis* Brongniart 1849

*Cladophlebis* cf. *denticulata* (Figure 4C)

Material: AG-M-90b and AG-M-91.

Description: Fragments of a second order pinna about 2.6 cm long and 1.8 cm wide. The pinnules have acute apices. The margins range from apparently smooth to slightly dentate. The average length of the pinnules is between 7.1–8.3 mm long and between 2.9–3.6 mm wide at the base. Pinnules present an opposite to slightly alternate distribution. There are 12 pinnules with different levels of preservation, which are inserted into the rachis at an angle of 61–75°. The rachis of the pinna can reach 1.1 mm wide. The venation of the pinnules is barely visible, showing a central vein that runs parallel to the sides of the pinnula and from which secondary nerves emerge towards the margin.

Gymnospermatophyta

Order Bennettitales Engler, 1892

Family incertae sedis

cf. *Pterophyllum* sp. (Figure 3B–D)

Material: AG-M-004a and AG-M-004b.

Description: The largest specimens are up to 5.6 cm long and 5 cm wide. Up to 28 leaflets are distributed alternately and separated by about 0.76 mm. The degree of insertion of the leaflets to the rachis varies between 55–88°.

Leaflets lanceolate-shaped, up to 25.1 mm long and 1.6–3.7 cm wide, with an acute apex and constant thickness throughout. Both the basal and apical leaflets are smaller than those in the central part of the leaf. The venation of the specimens is only observable in some parts due to preservation, consisting of a maximum to six veins that run parallel to each other without dividing at any time, and separated by approximately 0.2 mm.

cf. *Ptilophyllum acutifolium* (Figure 4F,G)

Material: AG-M-008; AG-M-009; AG-M-029a, b; AG-M-031; AG-M-039; AG-M-059; AG-M-060; AG-M-067a, b.

Description: These are incomplete specimens. The largest specimen fragment is 3.5 cm high and 4.8 cm wide. It preserves up to 20 leaflets that are distributed alternately separated by less than 0.5 mm from each other. The angle of insertion to the rachis is 51–60°.

Leaflets elongated and lanceolate with acute apex. Their width varies between 2.45 and 3.7 mm, and they reach a length of 34 mm in length. The venation is only visible in some fragments, with four to five veins that run parallel to the margin and each other with a separation of between 0.1 and 0.2 mm.

cf. *Ptilophyllum pecten* (Figure 4B,D)

Material: AG-M-021c

Description: The only specimen is an almost entire leaf, except for a small missing part in the apical area. It has a maximum length of 4.67 cm, and a maximum width in its middle part of 1.57 cm, narrowing towards both the base and apex. There are at least 26 leaflets attached to the rachis alternately to semi-alternately and they are 0.2–0.3 mm wide. The angle of insertion of the leaflets to the rachis varies between 53°–60°.

The leaflets are oblong in shape, with a rounded apex reaching a length of 11.4 mm and a width of 2.6–3.0 mm at their base, tapering towards the apex. The veins are only partially visible, with up to seven veins running parallel both to the margin and parallel to each other and separated by 0.1 mm.

cf. *Ptilophyllum pectiniformis* (Figure 4A)

Material: AG-M-001; AG-M-010; AG-M-011; AG-M-022 a, b; AG-M-024; AG-M-028; AG-M-032a, b, c; AG-M-033a, b; AG-M-035a, b; AG-M-036a, b; AG-M-047a, b, c, d, e; AG-M-048; AG-M-060; AG-M-066a, b, c, d, e.

Description: The specimens reach at least 10.36 cm long and up to 3.65 cm wide decreasing in width towards the apical zone. The specimens have up to 36 leaflets that are attached to the rachis with an angle of between 55–77° in a semi-alternate way. The width of the rachis varies between 0.2 and 1.0 mm wide.

The leaflets are lanceolate in shape, with a sharp and slightly asymmetric apex. They reach 25.4 mm in length and 4–5.3 mm in width in the basal area, narrowing towards the apex. The venation is only observable in some specimens, presenting between six and seven veins that run parallel both to the margin and parallel to each other and separated by about 0.1–0.2 mm.

cf. *Ptilophyllum* sp. (Figure 3A)

Material: AG-M-002; AG-M-005; AG-M-007; AG-M-012; AG-M-013; AG-M-014; AG-M-015; AG-M-016; AG-M-017; AG-M-018; AG-M-019; AG-M-020a, b, c; AG-M-023 a, b; AG-M-027a, b, c, d; AG-M-030a, b, c; AG-M-034a, b, c, d; AG-M-037a y b; AG-M-038 a, b, c, d; AG-M-040; AG-M-041a, b, c; AG-M-042 a, b, c;

AG-M-043a-l; AG-M-044a-l; AG-M-045a,b,c,d,e; AG-M-046; AG-M-049; AG-M-050; AG-M-051; AG-M-052; AG-M-053; AG-M-054; AG-M-055; AG-M-056; AG-M-057; AG-M-08; AG-M-061a, b; AG-M-062; AG-M-063; AG-M-064a, b; AG-M-065; AG-M-068; AG-M-069; AG-M-072; AG-M-073; AG-M-074; AG-M-075; AG-M-079; AG-M-082.

Description: Leaf fragments of different sizes, reaching up to 13 cm long and 7 cm wide, with several leaflets attached to the central rachis at angles between 48–82°, with alternate to semi-alternate distribution. The leaflets are mostly lanceolate-shaped reaching 4.6 cm in length, and they have an acute apex.

**Figure 3 biology-11-00273-f003:**
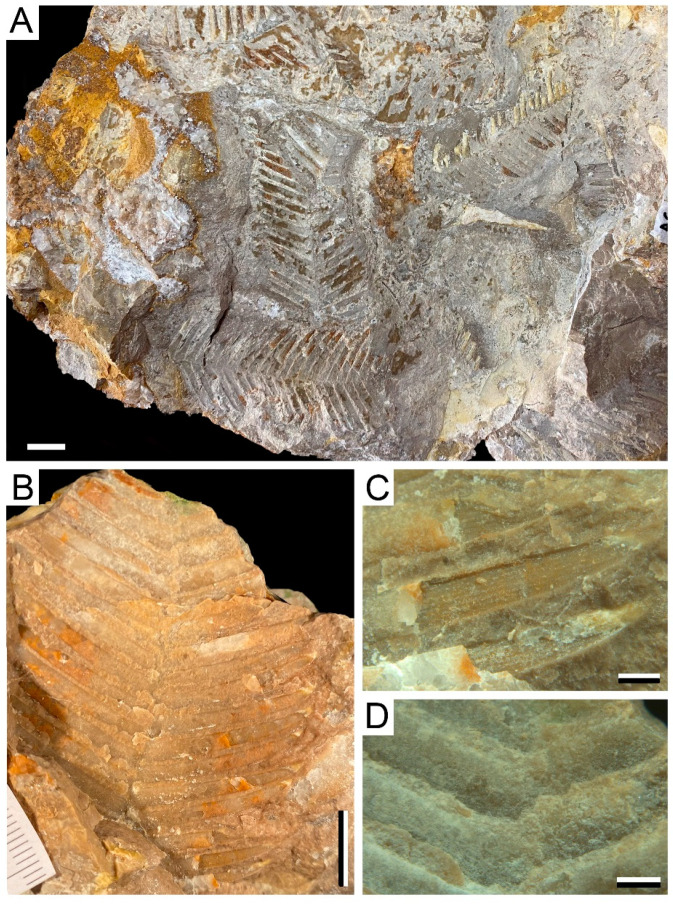
Plant remains from the Becerril section: (**A**). Example of orientation and preservation of macroflora remains in a rock-sample, showing different Bennettitalean-like specimens (cf. *Ptilophyllum* sp.), AG-M-44a-g. Bar = 1 cm; (**B**). cf. *Pterophyllum* sp., AG-M-004a. Bar = 1 cm; (**C**,**D**). Detail of venation and leaflets distribution of (**B**) cf. *Pterophyllum* sp., AG-M-004. Bar = 20 mm.

**Figure 4 biology-11-00273-f004:**
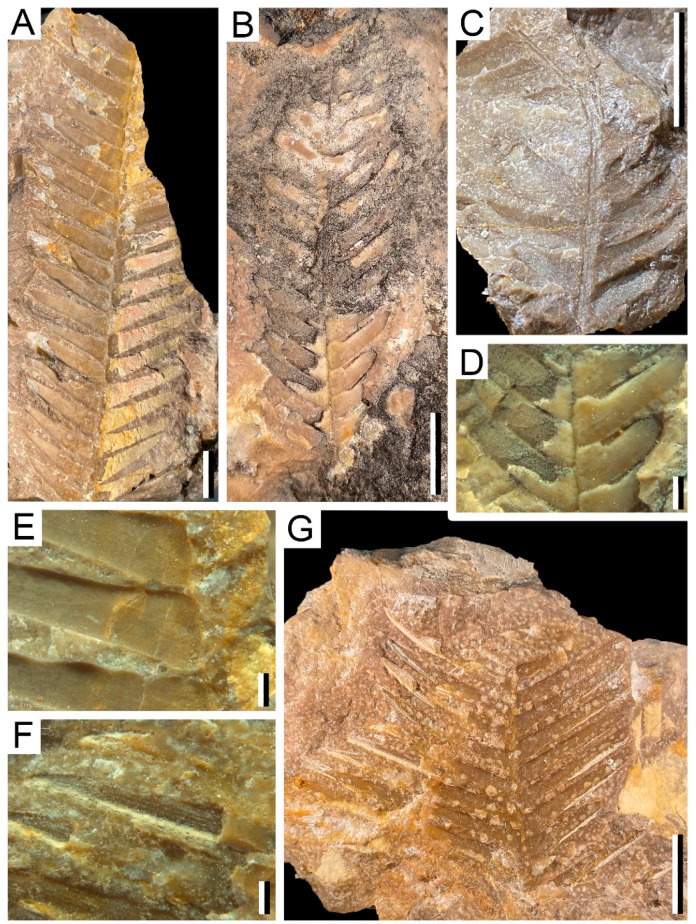
Plant remains from Becerril section: (**A**) cf. *Ptilophyllum pectiniformis*, AG-M-01b. Bar = 1 cm; (**B**) cf. *Ptilophyllum pecten*, AG-M-21c. Bar = 20 mm; (**C**) *Cladophlebis* cf. *denticulata*, AG-M-90b. Bar = 1 cm; (**D**) Detail of venation and leaflet insertion of (**B**) cf. *Ptilophyllum pecten*, AG-M-21c. Bar = 20 mm; (**E**) Detail of (**A**) cf. *Ptilophyllum pectiniformis*, AG-M-01b. Bar = 20 mm; (**F**) Detail of venation of (**G**) cf. *Ptilophyllum acutifolium*, AG-M-29b. Bar = 20 mm; (**G**) cf. *Ptilophyllum acutifolium*, AG-M-29b. Bar = 1 cm.

## 5. Discussion

### 5.1. Entomofauna

The discovery of a dragonfly so closely similar to the German *Cymatophlebia longialata* that we could not separate it from this species is rather surprising because the genus *Cymatophlebia* is currently only known from the Upper Jurassic and Lower Cretaceous of Europe and Central-Eastern Asia. The highest known diversity of this genus is currently known from the Bavarian Solnhofen lithographic limestone. The unique alternative (less probable, see above) would be an attribution to *Cymatophlebia kuempeli*, also from the Bavarian Solnhofen lithographic limestone. The Iberian Peninsula was a ‘continental’ island during the latest Jurassic, while the Bavarian dragonflies of the genus *Cymatophlebia* were living on a group of small atolls surrounded by a tropical sea. These large dragonflies possibly had significant capacity to migrate for rather long distances, as shown by their large cubito-anal areas of their hind wings (areas I and II vs. area III of Suarez-Tovar and Sarmiento [68]: Figure 1), similarly to their smaller modern relative *Hemianax ephippiger* that is able to migrate from Senegal to Northern Europe. The presence of ‘relay’ islands in France (outcrops of Cérin or Orbagnoux) renders the hypothesis of some entomofaunal exchanges between the Central Europe and Spain at that time.

The Aktassiidae are a rather small family of very large Odonata also recorded from the Upper Jurassic to Lower Cretaceous of Germany, UK, and Central-Eastern Asia. Some species had a rather wide distribution in China [69]. They had also broad cubito-anal areas in their hind wings, suggesting that they could accomplish significant migrations.

Unfortunately, we know nothing on the larvae of both Aktassiidae and Cymatophlebiidae. We can only suppose that they were living in freshwaters.

In general, the exceptional preservation of fossil insects in geothermally influenced environments makes their records particularly valuable. Modern-day hot spring systems can serve as local refugia for thermophilic animals and plants [8,12,13,14]. Thermal springs around the world have similar insect faunas; only four orders (Diptera, Coleoptera, Hemiptera, Odonata) are commonly represented, and each of these only by a handful of genera [70]. In a global summary, 38 species of Odonata were listed as permanent inhabitants of hot springs that can complete their life cycle in geothermal environments [8,9]. Due to the low amounts of individuals of only two species of dragonflies, this type of preservation in travertine supports the idea that these species were possibly abundant locally near geothermal springs, meaning that the specimens were buried where they had once lived.

Therefore, the here described dragonfly remains of the Aguilar Basin hot spring area represent obviously a former community of a species-poor(?) palaeobiocenosis living near geothermal habitats, such as an outflow stream or a pool accumulating geothermal waters from surrounding hot springs.

### 5.2. Palaeobotany: Palaeoenvironmental, Taphonomical, and Palaeogeographical Implications

The new (palaeo)flora found in Becerril is dominated by Bennettitales (98.5%), with ferns being a minority (1.5%). The 136 identified specimens only belong to 3 genera: *Ptilophyllum* (97%), *Pterophyllum* (1.5%), and *Cladophlebis* (1.5%). This floral composition implies an undoubtedly low diversity.

The composition of the floral assemblage of the travertines in the Villela fossil site [23] is very similar in diversity to the assemblage in the Becerril flora studied here (with the same three genera presented in both locations) except for the very doubtful presence in the Villela site of the taxa *Otozamites mundae* Morris and *Zamites pumilo* Sapota emend Barale and cf. *Pseudocycas* sp. However, the values of their composition are different, with the ferns representing 8.2% and the Bennettitales a 91.8% to the total in Villela [23]. On the other hand, in the Villela site the authors did not report the presence of insect remains. Nevertheless, the clear similarity between the two deposits is consistent with the interpretation that the Becerril deposit is an equivalent deposit to that of Villela. The absence in both sites of other typical taxa that are common components of other Upper Jurassic and Lower Cretaceous floral assemblages in the Euro-Sinian region *sensu* Vakhrameev [71] is striking.

Comparison of the flora studied with other plant assemblages of the Jurassic-Cretaceous boundary in the Iberian Peninsula is challenging since most of the few studies of this age focus on palynology, and only a few macrofloral studies of this age have been hitherto carried out. It is worth noting that both the macrofloral and microfloral-assemblages from the Early Berriasian deposits of Las Zabacheras in the Galve sub-basin (Northeastern Spain), are notably different from that described here (see Santos et al. [47]). However, the relative geographical and temporal proximity of both areas would suggest that the climatic conditions should be similar, only the genus *Cladophlebis* is common to the two sites, and while the flora studied is practically dominated by two genera of Bennettitales, Las Zabacheras macroflora was composed of a variety of ferns corresponding to different genera [47]. This change in the plant composition could be due to the different environmental contexts of both sites. Thus, the Becerril flora was located in a hot-spring with very specific environmental conditions, but the Zabacheras flora was related to a deltaic sedimentary environment near the coast [47,72]. This difference in the local/regional environment could clearly be a selective factor for the type of plants.

The plant remains of Becerril consist of centimetric-size fragments of leaves that do not retain their base but do retain the apex of the leaves in numerous specimens and with their leaflets in connection with the rachis. The fossil leaves do not present a defined orientation in the stratum (see Figure 3A), which points out that these remains would have suffered a short transport and not very intense up to the deposit area. These data would indicate that the studied material could be interpreted as autochthonous or (par)autochthonous material, growing in the sides of the hot-spring environment where they were finally buried.

The absence of other plant groups that are typical components of Jurassic-Cretaceous plant assemblages, such as conifers, cycadales, or ginkgoales, is striking. At this point, we agree with the interpretation of Diéguez et al. [23], in that these groups have a high preservation potential, so it seems undeniable that their absence in the fossil record preserved in this site is not related to taphonomic processes. In addition, both the arrangement and preservation of the fossil remains—including the insect wings—in this site suggests that it is a (para)autochthonous material, and a hydrodynamic bias seems also unlikely. During the Tithonian-Berriasian, the study area would be a shallow lake system, with development of wetlands and colonization by charophytes and subsequent desiccation [43], in which carbonate hydrothermal contributions would take place with the development of bacterial covering [20]. The abundance and low variety of cycadophytes present in the Becerril site, which are related to dry environments [23], would be consistent with an environment like the hot-spring related deposits where the fossils occur, which nowadays present conditions of marked environmental stress.

In a study of the stratigraphic record and the sedimentology of the area, Pujalte et al. [43] indicate that towards the NE and SE margins of the basin the freshwater marsh limestone lithosomes interdigitate until they wedge with clastic fluvial deposits. In addition, the presence of evaporitic deposits laterally correlated to these limestone lithosomes indicates that it would be a marsh area entirely surrounded by fluvial environments within an endorheic system with the presence of (semi) arid conditions, which would be in consonance with the presence of a xerophytic flora preserved in the limestone lithosome [23]. Moreover, the presence of different types of rhizolites in lateral facies correlated with these stratigraphic levels reflects changes in the substrate, from humidification conditions to desiccation, suggesting changes in the humidity/aridity conditions during the deposition of the Aguilar Fm. [63].

The most part of the taxa from the Aguilar Fm. corresponds to plants that present a wide range of environmental, geographical and temporal distribution in both hemispheres during the Jurassic and Cretaceous [71]. On the one hand, *Cladophlebis* is usually restricted to include Triassic to Cretaceous sterile ferns with a straight midvein and sparse dichotomous lateral veins. Due to the absence of reproductive characters this genus presents doubts about its assignation to some family of ferns [73], although some authors use to assign it to Osmundaceae [43,74,75,76]. Taking into account that the climate in the studied area was interpreted as (semi)arid [23,43] and that this family of ferns probably grew under warm, humid environmental conditions, during Mesozoic [47,75] it is probable that the hydrothermal activity provided a suitable warm and humid environment for this group of ferns to grow. According to Diéguez et al. [23] the presence of Bennettitales with additional characteristics as small leaves and thick leaflets, suggests arid conditions for the Villela flora. These characteristics are also common to the Becerril flora. Nevertheless, the interpretation may not be so simple because Bennettitales commonly grew in deltaic and “highly disturbed environments” under warm and subtropical conditions, similar to modern mangroves [77]. In addition, other authors suggested that the foliar characters of Bennettitales could be more related to adaptations to physiological drought, as they growth in low-pH soils, rather than a preference for truly dry conditions [78,79]. In extant hot springs the very low pH values are not unusual [10], indicating that although the general conditions of the area were (semi)arid, the hydrothermal environment could have provided warm and humid conditions which apparently were more suitable for the development of a plant community dominated by Bennettitales, as it is the case of the Becerril/Villela flora.

Regarding the records of Bennettitales present in different localities within the Euro-Sinian region in the western Tethys area, the evidence from the Kimmeridgian and Tithonian deposits of Brunn [80] and Solnhofen [81,82] in Germany, and those of Cerin [83] Orbagnoux [84], Canjuers [85], and Causse Méjean [86] in France show a dominance of the genus *Zamites* Brongniart (although not in Canjuers, where this taxon has not been recorded) in finely laminated deposits associated with coastal lagoon environments with continental input in coastal marine sedimentary environments. It is also noteworthy that, in all these sites, the Bennettitales leaf records are accompanied by remnants of coniferous branches bearing short, triangular, coriaceous leaves, which are strongly pressed to the axis, fundamentally belonging to genus *Brachyphyllum* Lindley and Hutton that is considered a plant linked to areas with notable environmental stress [86,87,88,89]. It should be noted that in the conditions of high environmental stress present during the deposit of travertines in the hot spring environment in Becerril, this type of coniferous remains has not been found so far in the studied deposit.

On the other hand, records from the base of the Cretaceous in Germany (Pott et al. [90] and references within) and England (see Watson and Alvin [90] and references within) show that the macrofossil assemblages of plants in marsh and lacustrine environments without marine influence present a wide variety of different plant groups, including several species of the bennettitalean genera *Pterophyllum*, *Ptilophyllum*, and *Zamites*. In the area of the southwest of Tethys, during the Jurassic, environments with high environmental stress such as small isolated volcanic islands, were namely colonized by Bennettitales. This fact has been evidenced, for example, in the macro-floristic assemblage found in the lagoon deposits of a small isolated volcanic island developed during the Middle Jurassic in the northern of Spain [50].

## 6. Conclusions

We have presented a new plant-insect assemblage from a Tithonian-Berriasian travertine site. Insect remains consist of wings of Odonata and represent the first record for the families Cymatophlebiidae and Aktassiidae for the Iberian Peninsula. The taphocoenosis of the flora was dominated by bennettitales (98.5%) including cf. *Pterophyllum* sp., *Ptilophyllum* cf. *pecten*, *Ptilophyllum* cf. *acutifolium*, *Ptilophyllum* cf. *pectiniformis* and cf. *Ptilophyllum* sp.; fern remains (*Cladophlebis* cf. *denticulata*) were also present in the flora (1.5%), this composition is quite different from other Tithonian-Berriasian floras from Spain. On the other hand, the presence of Odonata and this flora in (semi)arid conditions could be because the hot-spring provided a niche with constant conditions of warmth and humidity forming an “ecological oasis”. The presence of the Anisoptera *Cymatophlebia* cf. *longialata* is consistent with a higher Tithonian affinity; in addition, the tentative co-occurrence of this genus and species in Spain and Germany, with biological barriers during the J/K boundary, together with some anatomical features of *C*. *longialata*, suggests that this large dragonfly had capacity to migrate for rather long distances. As Odonata remain are in general extremely rare in travertine deposits, the present discoveries of two different species with several specimens strongly suggest that future research of fossil insects from the Aguilar Fm. should give abundant new fossils of other groups. These could result to be crucial for the study of the entomofaunas of the poorly known period of the Jurassic-Cretaceous boundary. A re-employment of the 19th century methods of collection of fossil arthropods in travertines, as developed by Munier-Chalmas (1872), will then be necessary.

## Figures and Tables

**Figure 1 biology-11-00273-f001:**
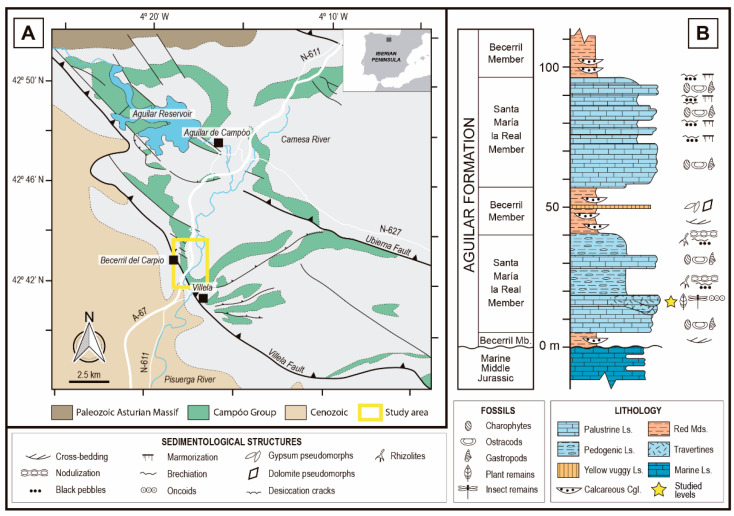
(**A**). Local geology of surroundings of Becerril del Carpio and study area, Campóo Group sediments highlighted. Modified from Diéguez et al. [23] and Pujalte et al. [43]. (**B**). Synthetized stratigraphic section from lower part of Aguilar Fm. in the Becerril del Carpio section. Ls.: Limestones, Cgl.: Conglomerates, Mds.: Mudstones. Modified from Hernández et al. [21].

**Table 1 biology-11-00273-t001:** Selection of worldwide hot-spring/travertine deposits containing fossil flora or fauna from Jurassic and Lower Cretaceous.

Location	Formation/Basin	Age	Geological Setting	Fossil Remains	References
Spain (Asturias)	Vega Formation (Asturian Basin)	Late Jurassic (early Kimmeridgian)	Oncoid-rich travertines associated with springs and ephemeral lakes near synsedimentary faults (lacustrine environment)	Roots, wood remains, charophytes, and ostracods	García-Ramos et al. (2010), Arenas et al. (2015), Lozano et al. (2016)
Spain (Palencia and Burgos provinces, Castilla y León)	Aguilar Formation (Aguilar Basin)	Jurassic-Cretaceous boundary (Tithonian-Berriasian)	Travertine facies that have been associated with hot springs probably linked to the Villela fault (fluvial/lacustrine environment)	Plant macro-remains (pteridophytes and Bennettitales), and insects (Odonata)	Ramírez del Pozo (1972), Schaaf, (1986), Schudack (1987), Hernández et al. (1998, 1999), Hernández (2000), Diéguez et al. (2009), This work
Argentina (Patagonia, Santa Cruz province)	Macizo del Deseado	Middle to Late Jurassic	Subaerial and sublacustrine hot spring environments (lacustrine environment)	Plants, arthropods, microbial, and fungal fossils	Echeveste (2005), Channig et al. (2007), Guido and Campbell (2009), Guido et al. (2010), Guido and Campbell (2011, 2012), García-Massini et al. (2012), Channing et al. (2011), Guido et al. (2019)
USA (Utah)	Navajo Sandstone Formation	Early Jurassic	Shallow spring-fed lakes ponded between aeolian dunes (desert oases environment)	Conifer trunks, ostracodes, charophytes, fish, mollusks, possible freshwater sponge, trace fossils, fragments of vascular plants	Parrish and Falcon-Lang (2007)
Canada (Nova Scotia)	Scots Bay Formation	Early Jurassic	Silica-rich hydrothermal springs and seeps around the floor of an aerobic lake	Algae, plants (wood remains, oncolites, charophytes, ostracods, gastropods, conchostraceans, fish bones	De Wet and Hubert (1989)
USA (Connecticut)	Coe’s Quarry (Hartford Basin)	Early Jurassic	Boiling hot spring setting characterized by micritic and banded travertines, cellular tufa, and abundant spherulites	Algae or bacteria evidence (stromatolitic structures)	Steinen et al. (1987)

## Data Availability

The specimens are housed in the collection of the “Las Loras UNESCO Geopark”, in Palencia, Spain.

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
