# Peer review of "Insect and Plant Diversity in Hot-Spring Ecosystems during the Jurassic-Cretaceous Boundary from Spain (Aguilar Fm., Palencia)"

_biology, 2022, doi:10.3390/biology11020273_

Round 1
Reviewer 1 Report
Dear Authors:
The authors should be congratulated for providing such a unique fossil assemblage. I really enjoyed reading the manuscript. I have listed a few inappropriate expressions, below. I am looking forward to seeing the manuscript in print.
Line 7, ‘[44, 45,46] Wile in’ missing a stop.
Line 9, Both ‘Fm.’ and ‘Formation’ are mixed-used in the manuscript. Please carefully check through the manuscript.
Lines 270, 283, 293, 304, 315, 328, Generic and specific names should be in italic.
Line 461, Delete ‘anatomical’.
Line 520, ‘Boundary’ should be changed to ‘boundary’.
Author Response
Please find attached the revised version of the paper entitled “Insect and plant diversity in hot-spring ecosystems during the Jurassic-Cretaceous boundary from Spain (Aguilar Fm., Palencia)” authored by Artai A. Santos, Andé Nel, Iván Rodríguez-Barreiro, Luis Miguel Sender, Torsten Wappler, and José Bienvenido Diez, and which we would like to be considered for publication in Biology. We are grateful with you and the referees for reviewing the manuscript and providing constructive suggestions. We have accepted almost all the corrections.
Reviewer 2 Report
This is a significant work describing the insect and plant diversity in hot spring ecosystem from Spain during the Jurassic- Cretaceous.
It’s a very interesting paper with a original and not published elsewhere data. This paper is properly organised, the title is informative and concise, the abstract is representative of the content. Authors based on the huge material and modern methods during their analysis.
The part of systematic palaeoentomology is prepared in a proper way according to the standard in this type of manuscript. Descriptions of all the samples (insects and plants) are complete.
Figures are good quality, useful, necessary and well labelled. Discussion is divided into two parts 5.1.-. Entomofauna and 5.2. -Palaeobotany: Palaeoenvironmental, taphonomical, and palaeogeographical implications what makes easy to understand for readers. Conclusions are supported by the data.
I tried to find any mistakes, errors or imprecisions in this paper but I couldn’t because it is a really good manuscript.
In my opinion, the overall quality of the work is high and I warmly recommend this article for publication in Biology in its present form.
Author Response

(The authors gave the same response as above.)

Reviewer 3 Report
The manuscript “ Insect and plant diversity in hot spring ecosystems during the 1 Jurassic-Cretaceous boundary from Spain (Aguilar Fm., Palencia) is a carefully elaborated work in each of its sections, well referenced and that provides important and innovative information about a Tithonian-Berriasian travertine site in the Iberian Peninsula. Given the unusual presence of the Tithonian-Berriasian floras in the Iberian Peninsula, the contribution of this work to their knowledge is manifest.
The first records for the Odonata families Cymatophlebiidae and Aktassiidae are an important contribution to the knowledge of the paleodiversity of animals and together with the plants and the biological and ecological interpretation of this ecosystems the objectives of this work are achieved.
The descriptions of the fossils are detailed and well discussed and the photographs that illustrate them of quality.
Author Response

(The authors gave the same response as above.)

Reviewer 4 Report
In this contribution, the authors present plant and insect remains preserved in a travertine deposit from the Jurassic-Cretaceous boundary of Spain. The scarcity of localities of these characteristics and age worldwide renders any findings of special significance. Two fossil dragonfly morphotypes based on a total of three specimens (two of them partial) and multiple plant remains largely belonging to bennettitaleans are described.
The manuscript is well written throughout. Figures are informative and of good quality (yet see below) and descriptions are detailed. The taphonomic reasoning is sound.
The only point I would like to highlight separately here --although please note I am not a dragonfly expert-- is that many of the characters that are used as diagnostic for Cymatophlebia and those which allow to identify a closer resemblance with C. longialata (e.g., no accessory cubito-antenodal crossveins between CuP-crossing and PsA, number of secondary antenodal crossveins between Axl and Ax2, number of cells in the subdiscoidal triangle, rows of cells in basal postdiscoidal areas and between longitudinal veins such as RP1 and RP2, between RP2 and IR2, or between IR2 and Rspl, or pseudo-veins in postdiscoidal area and between Mspl and MA) are challenging (if not impossible) to ascertain from the provided photographs, at least the ones that I had the chance to examine. The inset provided in Subfigure 1B is insufficient on that regard. This might be (partially?) due to the downsized quality of the photographs in the pdf provided for review, but it would help very much if interpretative drawings, at least of those areas with a higher number of diagnostic features, could be provided. I would recommend the authors to strongly consider this matter. If drawings cannot be provided, I would suggest providing higher magnification photographs of the wing areas with a greater diagnostic significance ensuring the best lighting as possible (see comment below), although even in these instances seeing details such as the ones noted above might prove difficult due to the type of preservation of the fossil.
Providing the best figuration possible in order to ensure a reliable and clear classification of the specimen is paramount since identifying C. longialata as the closest taxon to two of the Spanish specimens is used to reach inferences about the migratory ability of this particular species and suggest a likely Tithonian age for the studied site, as stated in the abstract and conclusions. On that regard, the tentative assignation of two of the studied specimens to C. longialata, as properly discussed in the manuscript, needs to be treated as such in both abstract and discussion in order to highlight the proper degree of certainty of the conclusions that derive from the taxonomic assignation of the fossil dragonflies. Abstract: "The presence of anisopterans of the genus Cymatophlebia tentatively assigned to C. longialata suggests a higher affinity for a Tithonian age of the studied site, and the anatomy and alleged palaeogeographical distribution of this species suggest the capacity to migrate for rather long distances." Conclusions: "The presence of the Anisoptera Cymatophlebia cf. longialata is consistent with a higher affinity for a Tithonian age of the studied site; in addition, the tentative co-occurrence of this species in Spain and Germany, with biological barriers during the J/K boundary, together with some anatomical features of C. longialata, suggests that this large dragonfly had the capacity to migrate for rather long distances." (edited parts in bold)
Aside from that, please add an abbreviation section in material and methods including all the abbreviations used in Fig. 1, as this information appears to be currently missing.
And one last comment regarding the wing photographs, in case it can be followed in the future by the authors and which does not substantially change the information provided by the present Fig. 1, is to check the photographic results using lower angles of light incidence, i.e., more parallel to the surface of the rock slabs (ideally using different light foci, perpendicular to the different sets of veins, e.g., longitudinal veins and crossveins). I suspect results could have been slightly to perhaps significantly better in terms of contrast for the provided samples (particularly in Fig. 1A and 1D), although it is always a balance and very variable depending on the sample’s preservation and relief. Thank you.
Other edits (largely minor) are given in the attached annotated pdf.

Author Response
Please find attached the revised version of the paper entitled “Insect and plant diversity in hot-spring ecosystems during the Jurassic-Cretaceous boundary from Spain (Aguilar Fm., Palencia)” authored by Artai A. Santos, Andé Nel, Iván Rodríguez-Barreiro, Luis Miguel Sender, Torsten Wappler, and José Bienvenido Diez, and which we would like to be considered for publication in Biology. We are grateful with you and the referees for reviewing the manuscript and providing constructive suggestions. We have accepted almost all the corrections.
The only point we respectfully disagree is modifying the brightness/contrast of the insect wing figures as suggested by the last reviewer. We tested different settings and we consider that the current figure is the best version to show the characters that we describe in the MS, the important taxonomic characters are also perfectly verifiable in the figures.
